# Accuracy of a Novel Preoperative Failure Risk Model for Debridement Antibiotics and Implant Retention (DAIR) in Acute Prosthetic Joint Infection

**DOI:** 10.3390/diagnostics12092097

**Published:** 2022-08-29

**Authors:** Ignacio Sancho, Iñaki Otermin-Maya, Jorge Gutiérrez-Dubois, Ignacio Aláez, Ángel Hidalgo-Ovejero, Julián Librero, María Eugenia Portillo

**Affiliations:** 1Department of Orthopaedics and Trauma Surgery, Hospital Reina Sofía, 31500 Tudela, Spain; 2Department of Orthopaedics and Trauma Surgery, Hospital Universitario de Navarra, 31008 Pamplona, Spain; 3Department of Internal Medicine, Hospital Universitario de Navarra, 31008 Pamplona, Spain; 4Instituto de Investigación Sanitaria de Navarra (IdiSNA), 31008 Pamplona, Spain; 5Navarrabiomed, Complejo Hospitalario de Navarra—UPNA, 31008 Pamplona, Spain; 6Department of Microbiology, Hospital Universitario de Navarra, 31008 Pamplona, Spain

**Keywords:** prosthetic joint infection, total hip arthroplasty, total knee arthroplasty, DAIR, debridement

## Abstract

Given the variable success of the debridement, antibiotics and implant retention (DAIR) procedure in patients with acute prosthetic joint infection (PJI), an accurate selection of candidates is critical. In this study, we set about calculating the predictive value of a novel algorithm for predicting outcome following DAIR developed by Shohat et al. Sixty-four patients who underwent debridement for (early and late) acute PJI in a tertiary-level university hospital were selected, and the aforementioned algorithm was retrospectively applied. Patients with model scores of 40–50%, 50–60%, 60–70%, 70–80% and 80–90% displayed success rates of 33.34%, 41.18%, 57.9%, 78.27% and 100%, respectively. The receiver operating characteristic curve showed an area under the curve of 0.69. The calibration intercept value was 0, and the calibration slope value was 1. Failure rates were significantly higher for the following variables: revision surgery (*p* = 0.012) index surgery for reasons other than osteoarthritis (*p* = 0.01), and C-reactive protein level >30 mg/L (*p* = 0.042). This analysis demonstrated that the Shohat algorithm is associated with an optimal calibration value and a moderate predictive value for failure of a DAIR procedure in patients with acute PJI. Its validation is recommended before it can be routinely applied in daily practice.

## 1. Introduction

Prosthetic joint infection (PJI) is a devastating complication that results in significant costs in terms not only of financial resources but also of morbidity and mortality. Despite a relatively low incidence (1–2% in primary surgery and up to 10% in revision surgery) [1,2], the exponential surge in the number of arthroplasties performed in the last few years means that the management of PJIs has become a serious public health problem in developed nations [3].

According to Zimmerli et al., when PJI manifests itself within three months from surgery, it should be classified as an acute PJI [4]. In patients with acute PJI and a stable implant, debridement, antibiotic therapy and implant retention (DAIR) is an attractive alternative [5,6]. Its theoretical advantages include less technical complexity and aggressiveness, faster recovery and lower associated costs as compared with prosthetic revision surgery.

However, DAIR is not without complications, which means that it should not be used indiscriminately. Its main disadvantage is its variable effect on infection control, with rates ranging from 16 to 88% according to the literature [7,8,9,10,11,12]. Such variability may be attributable to the heterogeneity of the studied series (most of them multicenter studies). The success rate is lower in late (also known as hematogenous) acute PJI (LAPJI) than in early acute PJI (EAPJI) [6,13,14]. Several studies have pointed out that patients undergoing one or several unsuccessful DAIR procedures may present with a poorer overall prognosis following two-stage revision [15,16], although no unanimity exists in this regard [17,18,19].

The success of DAIR depends on the combination of multiple interrelated factors. Some are related to the patient [5,9,20,21], or to the index surgery [21,22]; others to the time at which the debridement is performed [6,8,23,24], or to the surgical technique employed, or the analytical values (C-reactive protein (CRP), white blood cell (WBC) count, bacteriemia) [21], the microorganisms involved or their sensitivity to anti-biofilm antibiotics. However, given the heterogeneity of available studies and the variability in the results obtained, it is difficult to determine which of the failure predictors mentioned play a decisive role [25].

These risk factors have been the subject of analysis for some time, and several predictive preoperative tools have been developed for the risk of failure of DAIR. With these premises in mind, in 2015, Tornero et al. [9] developed a score called the Kidney, Liver, Index surgery, Cemented prosthesis and C-reactive protein value (KLIC) intended to predict failure among patients who underwent DAIR for EAPJI. The score was subsequently externally validated by other authors [23,25,26,27,28]. Sometime later, the European Study Group for Implant-Associated Infections (ESGIAI) developed another score specifically applicable to LAPJI [29]. More recently, Shohat et al. [21] published an algorithm applicable to both EAPJI and LAPJI based on machine-learning techniques. The authors analyzed over 1000 patients with acute PJI subjected to DAIR. The resulting model displayed a high discrimination power, with an area under the curve (AUC) of 0.74. The usefulness of this kind of tool is significant for clinicians and patients alike, allowing for the latter’s involvement in the decision-making process.

The specific epidemiology of different geographical areas and the differences in clinical practice between different hospitals make it necessary to validate these kinds of scores before their use can be generalized. Ideally, the effectiveness of these models should be evaluated according to the Transparent Reporting of a Multivariable Prediction Model for Individual Prognosis or Diagnosis (TRIPOD) guidelines for the study of prognostic models [30].

The purpose of this article was to analyze the accuracy of the score developed by Shohat et al. when applied to a series of patients with acute PJI subjected to a DAIR procedure in one single institution. This is, to the best of our knowledge, the first study to evaluate the validity of that algorithm.

## 2. Materials and Methods

### 2.1. Data Sources, Participants and Definitions

Institutional and ethical approval for using the data was obtained prior to commencing the study. During the period 2011–2019, a total of 8639 arthroplasties were recorded in a prospectively created database at the University Hospital of Navarre, a third-level university hospital located in Pamplona (Spain). Only total hip arthroplasties, total knee arthroplasties, hip revision surgeries and knee revision surgeries for which complete data were available were included. The definition of PJI published in 2011 by the Musculoskeletal Infection Society (MSIS) was used [31]. Only acute infections subjected to a DAIR procedure were included. EAPJI was defined as an infection arising within three months from the surgery and treated during that period. LAPJI was defined as a late acute infection arising beyond three months from the index surgery, characterized by an abrupt onset of symptoms over a joint with a prior adequate status.

A total of 64 PJI cases subjected to a DAIR procedure were identified (55 EAPJIs and 9 LAPJIs), which complied with all the above-mentioned criteria. Patients who did not comply with the MSIS infection definition, those with (primary and revision) tumor arthroplasties, those not followed-up for at least 24 months and those where data were incomplete were all excluded. To optimize consistency with Shohat et al.’s criteria, only patients where DAIR was performed within 3 months from onset of symptoms for EAPJI and within 3 weeks from onset of symptoms for LAPJI were recruited (Figure 1).

### 2.2. Treatments Administered

Most of the members of the debridement surgical team (i.e., joint prosthetic surgeons from the knee and the hip units) had also been involved in the index surgery. The surgical team remained unchanged throughout the study period.

Whenever possible, preoperative synovial fluid samples were taken through arthrocentesis, but in no case was debridement held up while waiting for the culture results to become available. Following the previous surgical approach, collections were thoroughly drained, followed by a radical debridement of all nonviable tissues. A profuse irrigation was conducted with at least 6 L of saline. Surgeons were free to decide on the use of other antiseptic solutions and on the exchange of the modular components (femoral head prosthesis and inserts). Samples of the synovial fluid and the affected tissues were sent to the laboratory for microbiological analysis, and, in some cases, the explanted modular components were sonicated. All DAIR procedures included were open-surgery procedures.

The hospital’s Internal Medicine Department, which specializes in treating prosthetic infections, was responsible for defining the kind of antimicrobial protocol to be used. Initially, broad-spectrum intravenous antibiotic therapy was administered if the causative organism was unknown. This was followed by targeted oral antibiotic treatment for 6–12 weeks. Unless contraindicated, rifampicin was always present in the postoperative regimen; the treatment always comprised a combination of drugs. In none of the DAIR cases recruited were carriers or local antibiotics applied.

### 2.3. Outcome-Related Definitions

DAIR was considered to have failed when the patient required prosthesis removal or when they passed away as a result of PJI during the follow-up period, when they required antibiotic suppression therapy or in cases of reinfection.

### 2.4. Predictors

A database was created containing a total of 57 predictors for each one of the 64 patients undergoing DAIR.

Predictors related to the index surgery included indication (osteoarthritis (OA)/other), involved joint (knee/hip), type of surgery (primary/revision) and use of cement (yes/no). The cement used in all cases was antibiotic-free. Cases with just one cemented prosthetic component were regarded as cemented.

Patient-related predictors included age, sex, body mass index (BMI), preoperative anesthetic risk as defined in the American Society of Anesthesiologists (ASA) classification, presence of cardiovascular disease, respiratory disease, hypertension, diabetes mellitus, gastropathy, immunodepression, malignancy, chronic renal disease, rheumatism, liver disease, chronic use of anticoagulant or antiaggregant medication, active smoking and alcohol abuse at the time of the procedure.

Predictors related to the clinical manifestations observed included the presence of surgical wound leakage, hematoma or surgical wound infection as evidenced by positive culture, fistula or fever >38°. Laboratory parameters tested comprised plasma CRP levels (in mg/L) and WBC count. The measurements taken closest to the debridement surgery were always those used for the analysis.

Debridement-related predictors included time (in days) from the index surgery in the case of EAPJI and time (in days) from the onset of symptoms for LAPJI, and whether the modular components were revised or not.

Microbiological predictors included the results of blood cultures and of surgical sample cultures (type of organism and percentage of positive samples). Cases where more than two samples were isolated were regarded as polymicrobial; *S.lugdunensis* was classified as a separate category from coagulase-negative staphylococcus (CoNS) because of its different pathogenic behavior.

### 2.5. Statistical Analysis

An evaluation was made of the association between the potential predictive variables and the outcome obtained (success or failure of the DAIR procedure). Categorical variables were expressed as absolute frequencies and percentages, and were compared using the chi-squared test or Fisher’s exact test. In continuous variables, the France–Shapiro test was used to assess normality. If the variable had a normal distribution, the contrast between success and failure of the DAIR was analyzed using Student’s t-test; otherwise, the Mann–Whitney U test was applied. However, in order to compare the continuous variables’ distribution in our sample with the one used in the generation of Shohat’s score, since there was not enough information to perform a nonparametric test, Student’s t-tests were always used. Statistical significance was set at *p* < 0.05.

Some continuous distribution parameters were analyzed by means of a categorization. To this end, the points at which the contrast between the distributions of the variables for success vs. failure cases was maximal were adopted as cut-off points. This procedure resulted in the inclusion of the following predictors: age >70 years, BMI >30, ASA score >2, CRP >30 mg/L, more than 25 days to DAIR for EAPJI, and more than 7 days from onset of symptoms for LAPJI.

In order to ensure the applicability of this validation analysis, an effort was made to process the data as similarly as possible to Shohat’s study [21]. It should be mentioned that although Shohat et al. make a distinction between ischemic heart disease and heart failure, we grouped both conditions under the heart disease heading. Moreover, in our analysis, the liver disease category not only includes cirrhosis but also other severe liver conditions, and rheumatism does not just include rheumatoid arthritis but also psoriatic arthritis with a concomitant arthropathy. The success and failure criteria and the definition of prosthetic infection are identical to those used in the model to be validated.

The Shohat score (probability of success of DAIR) was calculated for each one of the 64 patients using the online software described by the authors in their article [32] (Appendix A).

### 2.6. Model Performance

The performance dimensions analyzed were discrimination and calibration [30]. The first was calculated by means of the area under the receiver operating characteristic curve (ROC), which may be interpreted as the probability of correctly classifying, or discriminating between, a couple of randomly selected patients, one with and the other without the outcome to be analyzed, i.e., a successful DAIR procedure.

The concept of calibration reflects a coincidence between the probabilities predicted by the model and the rates observed for the event of interest [33]. For a score to be applicable to clinical practice, it is not enough for the AUC to be within acceptable values; the calibration must also be correct [34]. Calibration, particularly when small cohorts are concerned, should be determined based on calibration intercept values and calibration slope values and of Hosmer–Lemeshow’s goodness of fit test. The target value of the calibration intercept is 0, with negative values indicating an overestimation of the predicted risk and positive ones an underestimation of the risk. The calibration slope, which evaluates the spread of predicted risks, should ideally have a value of 1. Slope values <1 suggest extreme predicted risks (too high for high-risk patients and too low for low-risk patients).

Lastly, the optimal cut-off point was calculated based on the Youden index (YI). The ROC curve takes into consideration all the consecutive cut-off values to define a high-risk group vs. a low-risk group. The YI can be defined as the sum of sensitivity and specificity −1 and reaches its maximum value at the left upper corner of the ROC curve [33]. The R v.4.1.0 and R Core Team software packages (2021) were used for the analysis [35].

## 3. Results

### 3.1. Univariate Analysis

The analysis included a total of 64 patients subjected to DAIR (55 cases of EAPJI and 9 cases of LAPJI), with the overall success rate of the DAIR procedure at 60.9% (95% CI) (0.48, 0.73).

Half of the patients (n = 32) received a knee prosthesis and the other half a hip arthroplasty. Fifty-two were primary surgeries, while twelve were revisions (18.8%). The most frequent indication of primary surgery was OA (68.8% of cases). Implants were cemented in 54.7% of cases. As regards patient demographics and comorbidities, mean age was 66.4 years, and 68.8% were male. BMI was 31.7, and 87% of patients were preoperatively classified as ASA II or III, with only 1.6% being assigned to the ASA IV category.

As far as laboratory findings were concerned, the mean plasma CRP value was 108 mg/L, and the mean WBC count was 9.2 × 10^9^/L.

The debridement procedure was carried out, on average, 35.1 days from the index surgery in patients with EAPJI and 15.3 days from the onset of symptoms in those with LAPJI. Modular components were revised in 30 of the 64 cases (46%).

Microbiology analyses managed to identify the causative organism in 93% of cases. Most infections had been caused by CoNS or methicillin-susceptible *Staphylococcus aureus* (MSSA) (26% were caused by both microorganisms). Up to 15% of infections were polymicrobial.

### 3.2. Bivariate Analysis

Table 1 shows the distribution of the studied variables according to the outcome of the DAIR procedure.

Predictors such as previous thromboembolic disease, mental illness, implanted pacemaker, acquired human immunodeficiency virus and previous prosthetic infection were left out of the analysis given their low frequencies. Similarly, cases where the DAIR procedure was performed by a surgical team different from the usual one and cases where the procedure was treated as an emergency were duly recorded. However, as their frequencies were extremely low, they were eventually left out of the analysis.

### 3.3. Differences between the Cohorts

The most significant difference had to do with sample size. Shohat’s population included 1174 patients with PJI recruited from several hospitals. Our cohort was made up of 64 patients, all of them treated in a single institution and by the same surgical team.

The success rate of the DAIR procedure in our cohort (60.9%; 95% CI: 0.48, 0.73) was lower than that reported by Shohat et al. (65.5%), although the difference was not significant (*p* = 0.4551).

Table 2 shows a comparison of the two cohorts according to the main predictors considered. Note the significantly higher frequency in Shohat’s cohort of cemented arthroplasties, alcoholism, LAPJIs and early debridement. Moreover, Shohat et al. found higher WBC counts, more positive blood cultures and more polymicrobial cultures.

In contrast, in our cohort arthroplasties performed for reasons other than OA, smoking and rheumatism were significantly more frequent. As regards the symptoms, more wound infections, fistulas and cases of fever were observed.

### 3.4. Algorithm Values

A comparison between the failure rates predicted by the score and those actually observed in our cohort is presented in Table 3. The table shows that there were no cases with a success score <40% in our series, with the majority of cases exhibiting success rates between 40 and 80%.

Application of Shohat’s score to all 64 patients yielded an AUC of 0.69, Figure 2. A 50% cut-off point in the score value showed sensitivity = 0.82, specificity = 0.44, and negative predictive value = 0.611.

Calibration of the algorithm was analyzed using Hosmer–Lemeshow’s goodness of fit test, which indicated poor fit and calibration (*p* < 0.05). The calibration intercept was 0, and the calibration slope was 1 (Figure 3).

## 4. Discussion

Although DAIR is currently a standard strategy for the management of acute PJI, its success rates are somewhat variable according to the reviewed literature. Failure risk factors tend to vary significantly across the published studies, which makes it difficult to establish their individual significance as predictors of failure.

The recently published Shohat score [21] uses many of these predictors to estimate the likelihood that a patient undergoing a DAIR procedure may be cured by the technique. The main purpose of this study is to provide external validation to the Shohat algorithm, as, according to the authors, it is essential to validate this kind of tool before it can be implemented in clinical practice.

Application of the algorithm to the patients in this study evidenced the perfect calibration of the model. The calibration intercept was 0, which is its target value (negative values indicating an overestimation of the predicted risk and positive ones indicating an underestimation of the risk). The calibration slope was also at its target value, i.e., 1, which shows an ideal spread of the predicted risk without over- or underestimating the risk that the technique might fail [33]. As shown in Figure 3, the fact that both values were ideal did not guarantee that the calibration curve would lie on the diagonal 45° line. For this to happen, a much larger patient sample would have been necessary. It is precisely for this reason that the poor fit-calibration result obtained from Hosmer–Lemeshow’s goodness of fit test is not contradictory. A high calibration value is essential for the model to be applicable to clinical decision making.

An evaluation of the discriminatory power of the Shohat score for our cohort showed a moderate predictive power (AUC = 0.69) for detecting the success of the DAIR procedure, which is lower than that reported by the original authors (AUC = 0.74). In our opinion, there are multiple causes behind this underperformance, including differences related to local epidemiology and to patient characteristics.

An analysis of the success rate of the DAIR technique yielded a lower score in our series (60.9%) than in Shohat’s (65.5%), although the difference was not statistically significant. The difference could be attributed to the fact that our series included a higher number of arthroplasties performed for reasons other than OA and higher rates of rheumatic patients, patients with pulmonary disease and smokers. Moreover, time to debridement was significantly longer in our series than in Shohat’s, both in cases of EAPJI and LAPJI, although a subanalysis of the curves corresponding to these variables and of the cut-off points did not indicate that the longer time to DAIR resulted in an excessively higher failure rate. Although the DAIR procedure should be performed as early as possible, an increasing number of authors have reported longer times to DAIR [24]. In addition, in our series, we were not able to establish a date after which debridement should be advised against. Possibly, the inoculum size, the type of microorganism and a handful of host-related factors are possibly decisive for the maturation of the biofilm, which means that the outcome of the DAIR procedure is not only a question of time. On the contrary, patients in this study were also younger than those of Shohat et al., with lower rates of cemented prostheses and a lower incidence of LAPJI, which have been shown to be factors conducive to higher rates of success in the context of DAIR.

The results obtained from our cohort have identified the following predictors of failure: an indication different from OA in the index surgery, revision surgery and CRP levels >30 mg/L. However, some of the variables often cited in the literature as significant predictors of failure were not found to be such in our study. These include the revision of mobile components, age, microorganisms and the already mentioned time to debridement. It should be mentioned that in spite of the low proportion of liners revised (53.1%), the success rate of the DAIR procedure in our series fell within the range reported by other authors. This could be attributable to the fact that the majority (84%) of liners of the hip prostheses used in this cohort were made of ceramics rather than polyethylene.

Our study presents a series of limitations. The most significant of them is the retrospective nature of the analysis, which made it necessary to leave out cases for which data was incomplete. Secondly, although this is one of the largest studies analyzing the failure of DAIR procedures, the size of the sample is still rather low. This is the reason why our comparison between early and late acute infections was nondiscriminatory, given that we only had nine cases of the latter.

The study does have, however, a number of strengths. One of them is the administration of the same treatment to all patients and the involvement of the same surgical team in all cases. With most studies on this subject being multicenter, they tend to be more heterogeneous. Moreover, the study was thoroughly consistent in terms of the criteria, cut-off points and definition applied. The study that developed the model to be validated was at all times taken as a reference.

A predictive model for the failure of the DAIR procedure based on risk factors such as those analyzed here constitutes a useful decision-making tool, particularly for doubtful cases. DAIR is a useful procedure for patients presenting with just a few of the risk factors analyzed and a high predicted success probability as it may avert the risk of having to remove the implant, with the potential morbidity and mortality that may result from that procedure. Similarly, a revision surgery should be indicated in cases with a low predicted success rate and a high number of predictors of failure. This may indeed speed up recovery and reduce the number of surgeries.

In our opinion, given the characteristics of the patients involved and the pros and cons of the DAIR procedure, an effort may be needed to minimize the number of false negatives, i.e., those patients likely to benefit from a DAIR procedure but not subjected to one because the algorithm suggests a high failure rate. In other words, what is needed is a sensitive algorithm endowed with a high negative predictive value. Analyzing the cut-off points on the ROC curve, a Shohat score of 50% provides a sensitivity of 0.82 with a negative predictive value of 0.61. In our opinion, this value should be taken into consideration during the decision-making process.

New tools may be developed in the future based on machine learning and next-generation big data technologies that may contribute to detecting ideal candidates for the DAIR procedure.

## 5. Conclusions

According to the findings from our study, performance of a DAIR procedure should be seriously meditated in patients with an acutely infected primary arthroplasty implanted for an indication different from OA or a revision arthroplasty, or in those with plasma CRP levels above 30 mg/L.

Shohat’s score is associated with moderate accuracy and optimal calibration when predicting the success of a DAIR procedure in patients with acute hip and knee prosthetic infections. The algorithm should be individually validated prior to being routinely applied to other cohorts in daily practice.

## Figures and Tables

**Figure 1 diagnostics-12-02097-f001:**
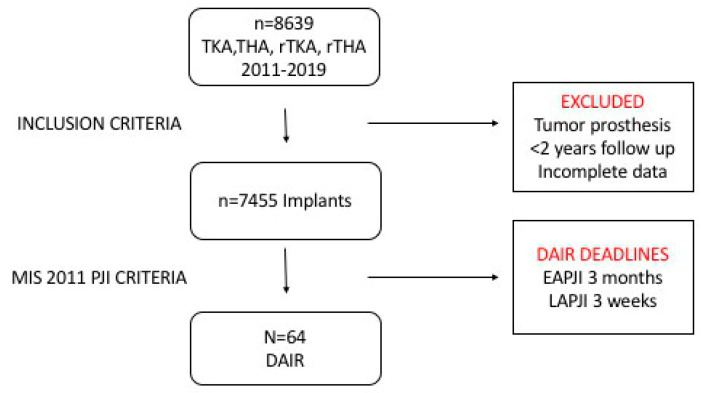
Patient recruitment flowchart. DAIR: Debridement, antibiotics and implant retention, EAPJI: Early acute prosthetic joint infection, LAPJI: Late acute prosthetic joint infection, PJI: Prosthetic joint infection, THA: Total hip arthroplasty, TKA: Total knee arthroplasty, rTHA: Revision total hip arthroplasty, rTKA: Revision total knee arthroplasty.

**Figure 2 diagnostics-12-02097-f002:**
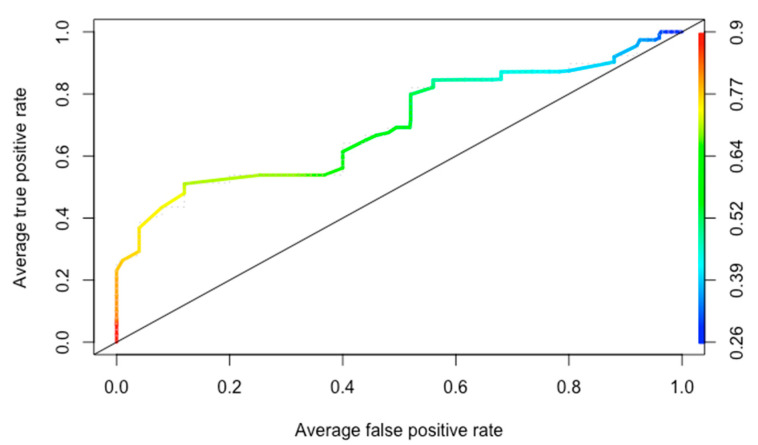
Receiver operating characteristic curve.

**Figure 3 diagnostics-12-02097-f003:**
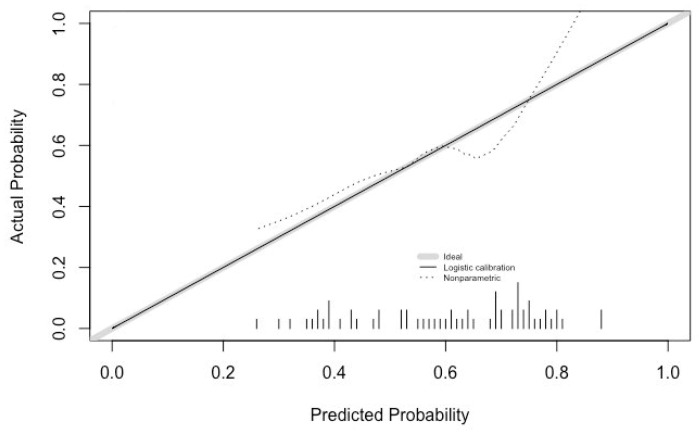
Calibration analysis.

**Table 1 diagnostics-12-02097-t001:** Predictors and frequencies according to the final outcome (DAIR success vs. failure).

	Success n = 39	Failure n = 25	*p* Value
**Index surgery, n (%)**			
Joint = Knee	20 (51.3)	12 (48.0)	1.000
Type of surgery = Revision	3 (7.7)	9 (36.0)	0.012
Cemented prosthesis	22 (56.4)	13 (52.0)	0.930
Indication = Other than OA	7 (17.9)	13 (52)	0.010
**Patient characteristics**			
Age (median (IQR))	68.00 [58.50, 76.50]	66.00 [60.00, 76.00]	0.984
Age > 70, n (%)	16 (41.0)	9 (36.0)	0.889
Gender = M, n (%)	13 (33.3)	7 (28.0)	0.863
BMI (median (IQR))	31.00 [26.50, 34.50]	33.00 [29.00, 36.00]	0.116
BMI > 30, n (%)	23 (59.0)	18 (72.0)	0.428
ASA > 2 (%)	12 (30.8)	11 (44.0)	0.418
Cardiovascular disease, n (%)	14 (35.9)	8 (32.0)	0.960
Respiratory disease, n (%)	8 (20.5)	8 (32.0)	0.460
Hypertension, n (%)	22 (56.4)	17 (68.0)	0.506
Diabetes, n (%)	10 (25.6)	8 (32.0)	0.789
Gastropathy, n (%)	8 (20.5)	4 (16.0)	0.902
Immunodepression, n (%)	4 (10.3)	0 (0.0)	0.261
Malignancy, n (%)	6 (15.4)	4 (16.0)	1.000
Chronic renal failure, n (%)	4 (10.3)	3 (12.0)	1.000
Rheumatism, n (%)	10 (25.6)	3 (12.0)	0.315
Hepatopathy, n (%)	0 (0.0)	2 (8.0)	0.290
Anticoagulant drugs, n (%)	6 (15.4)	2 (8.0)	0.628
Antiplatelet drugs, n (%)	8 (20.5)	8 (32.0)	0.460
Smoking, n (%)	17 (43.6)	13 (52.0)	0.688
Alcohol, n (%)	10 (25.6)	3 (12.0)	0.315
**Clinical findings, n (%)**			
Wound drainage	23 (59.0)	14 (56.0)	1.000
Hematoma	10 (25.6)	8 (32.0)	0.789
Skin infection	20 (51.3)	9 (36.0)	0.347
Fistula	13 (33.3)	11 (44.0)	0.552
Fever	11 (28.2)	11 (44.0)	0.304
**Laboratory**			
Serum CRP (median (IQR))	59 [21.5, 97]	85 [59.1, 180]	0.053
Serum CPR (mg/L) > 30, n (%)	26 (66.7)	19 (76)	0.042
WBC × 10^9^/L (median (IQR))	7.30 [6.10, 10.20]	9.80 [6.80, 12.10]	0.180
**DAIR characteristics**			
Time from index surgery to DAIR (median (IQR))	29.50 [22.50, 41.00]	29.00 [24.00, 50.00]	0.993
Time from index surgery to DAIR >25, n (%)	24 (70.6)	13 (61.9)	0.711
Time to onset of symptoms (median (IQR))	7.00 [3.75, 12.25]	20.00 [8.00, 34.00]	0.134
Time to onset of symptoms >7, n (%)	6 (50)	7 (77.8)	0.399
No liner exchange (%)	15 (38.5)	15 (60.0)	0.153
**PJI**			
Type of PJI = Late acute (%)	5 (12.8)	4 (16.0)	1.000
Positive blood culture 10^9^/L (%)	2 (5.1)	2 (8.0)	1.000
% of positive cultures (median (IQR))	66.66 [33.33, 100]	66.66 [40, 100]	0.251
Shohat (mean (SD))	67.74 (9.93)	61.21 (8.69)	0.009
**Microorganisms**			
CoNS	12 (30.8)	5 (20.0)	0.3931
*Corynebacterium* spp.	1 (2.6)	0 (0)	1.000
*E. faecalis*	0 (0)	1 (4)	0.3906
*L. monocytogenes*	1 (2.6)	0 (0)	1.000
*C. acnés*	1 (2.6)	1 (4)	1
*P.aeruginosa*	1 (2.6)	1 (4)	1
*S. dysgalactiae*	2 (5.1)	1 (4)	1
*S. lugdunensis*	2 (5.1)	0 (0)	0.5164
*S. pneumoniae*	1 (2.6)	0 (0)	1.000
*S. viridans*	1 (2.6)	0 (0)	1.000
MRSA	1 (2.6)	1 (4)	1
MSSA	8 (20.5)	9 (36)	0.2465
Polymicrobial	5 (18.82)	5 (20)	0.494
Culture negative	3 (7.69)	1 (4)	1

ASA: American Society of Anesthesiologists; BMI: body mass index; CoNS: coagulase-negative staphylococcus; CRP: c-reactive protein; DAIR: debridement antibiotics and implant retention; IQR: interquartile range; M: male; MRSA: methicillin-resistant staphylococcus aureus; MSSA: methicillin-sensitive staphylococcus aureus; OA: osteoarthritis; PJI: prosthetic joint infection; SD: standard deviation; WBC: white blood cell.

**Table 2 diagnostics-12-02097-t002:** Comparison between cohorts.

	This Study	Shohat et al.	*p* Value
Cases (n)	64	1174	
DAIR success (%)	60.9	65.5	0.4551
Knee joint (%)	50	51	0.87
Revision (%)	18	21	0.56
Indication other than OA (%)	31	16	<0.001
Cemented prosthesis (%)	54.7	70	<0.001
Modular parts unchanged (%)	46.9	50.25	0.6027
Age mean (SD)	66.4 (12.2)	70.1 (11.91)	0.0207
Pulmonary disease (%)	25	14.9	0.03
Diabetes (%)	28.1	20.6	0.15
BMI (mean, SD)	31.7 (5.9)	30.7 (6.54)	0.1935
Malignancy (%)	15.6	11.8	0.36
Kidney disease (%)	10.9	7.6	0.20
Rheumatism (%)	20.3	7.2	<0.01
Anticoagulants (%)	17	12.5	0.34
Smoking (%)	46.9	25	<0.01
Alcoholism (%)	20.3	36.6	0.01
Skin infection (%)	45.3	27.3	<0.01
Fistula (%)	37.5	25	0.02
Fever (%)	34.4	22.3	0.02
Type of PJI = Late acute (%)	14.1	32.7	<0.01
Days to DAIR, mean (SD)	35.1 (17.6)	14.2 (14.4)	<0.01
Days to onset of symptoms, mean (SD)	15.3 (13.7)	2.23 (6.2)	<0.01
Serum CRP mean (SD) (mg/L)	108 (101)	126.7 (9.2)	0.14
Plasma WBC count × 10^9^/L, mean (SD)	9.2 (4.1)	11.36 (4.1)	<0.01
Positive blood cultures (%)	6.2	22.5	<0.01
Polymicrobial (%)	15	28	0.02

BMI: body mass index; CRP: C-reactive protein; DAIR: debridement antibiotics and implant retention; OA: osteoarthritis; PJI: prosthetic joint infection; WBC: white blood cell.

**Table 3 diagnostics-12-02097-t003:** Prediction accuracy. Mean predicted success vs. mean observed success.

Shohat Ranks (%) ^a^	10–20	20–30	30–40	40–50	50–60	60–70	70–80	80–90	90–100
Cases (n)	0	0	0	3	17	19	23	2	0
Failure (n)	0	0	0	2	10	8	5	0	0
Mean predicted success (Shohat)	0	26	36	46	55	65	75	85	92
Mean observed success	NA	NA	NA	33.34	41.18	57.9	78.27	100	NA

NA: not applicable; (^a^) probability of success.

## Data Availability

Datasets and R software package files are available from the corresponding author upon request.

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
