# Peer review of "Accuracy of a Novel Preoperative Failure Risk Model for Debridement Antibiotics and Implant Retention (DAIR) in Acute Prosthetic Joint Infection"

_diagnostics, 2022, doi:10.3390/diagnostics12092097_

Round 1
Reviewer 1 Report
This study looks at the DAIR method for early treatment of acute postoperative infections in total joint arthroplasty. The important take home is the fact that infections where the CRP>30, diagnosis is not osteoarthritis and the procedure was other than a primary case, the results are significantly poorer. These conclusions merit publication with some modest improvements.
In general, this is an interesting paper and has numbers that are high enough for viable conclusions. I am not an expert at machine learning, and defer this to others. What you have seems fine to me.
I have only one suggestion for the authors to consider. The authors assess the many variables of comorbidity that may affect outcome but then compare them in unitary fashion. The finding is negligible in table 1, so they just don’t know the effect.
The authors should consider using one of the comorbidity tools like the Charlson Comorbidity Index. This index has been attempted in total joint data base reviews not showing statistical difference, likely due to the intention to treat bias. However, when you have a series of infected cases, the intention to treat bias is no longer an issue. You are stuck with a difficult case and you must treat it. This WAS shown using the CCI with hip fractures in the aged.
What a comorbidity index allows you to do is to factor a healthy patient with minimal comorbidities versus a patient who has lots of comorbidities. This is not happening in table 1 if comorbidities are considered as single factors. Charlson was able to show that cancer patients with a CCI of greater than 5 had a 14 fold higher incidence of death from cancer at one year. The reason is patient immune capability. If you have good data, the CCI can be very simple to calculate that takes a few moments. Check out the CCI tools on Google. That finding would increase the importance of this paper dramatically, as you have addressed one of the key elements determining the outcome of infection. I am anxious to see what you might find.
In my opinion, you are correct; the negative predictive value is the other key decision maker. It should be in the 80-90% range. We can deal with the exigencies of infected or probably infected but what if we just don’t quite know. We need a good tool.
The other factor that may be predicted in the future is the ability of PCR panels to identify more bacteria than culture and tissue biopsy. The 15% polybiome infections, are not unexpected and probably is comparable to other studies. In open wounds I know this number goes to 50-60%. You may disclose this fact. The problem with PCR for your paper is you have some unusual bacteria. PCR matrix panels are limited to only the most likely, again not helping the negative predictive value. You might tie this fact into your discussion as I find your data pretty compelling.
Line113 should be ‘recruited’
Line 129 suggest ‘femoral ball’
Line 223 is MSSA, not MRSA.
With these minor considerations I would recommend publication. Good luck!
Author Response
The authors thank the reviewers for carefully considering the manuscript and acknowledging our constructive engagement.
Changes are highlighted in red and some details are provided below.

Reviewer 2 Report
diagnostics-1846149
This is a very interesting study about PJI diagnosis and machine-learning. Therefore, it really fits with the scope of the journal.
In general, the manuscript is very well written and sound. It is easy to read and well structured.
I would suggest revising to a more “easy” English (for instance “complex constellation of interrelated factors”
The results answer the purpose of the study and the conclusion is clear
I have some suggestions.
Abstract
Fine
Introduction
Line 44: refer to “faster recovery” or “better functional outcomes” since most of patients that need a prosthesis are retired.
M&M
How infection was ruled out in revisions?
DAIR procedure include exchange of the non-fixed components. Why the surgeon decided in some cases not to revise them?
Results
Some 30% of cases had a prosthesis for another reason than OA. Which are those diagnosis?
50% of cases had not liner/ polyethylene exchange. This is my major concern of the present study as there is quite good evidence that it is a risk factor for DAIR failure. It should be stated as a limitation of the study
Discussion
Fine
Conclusion
Lines 377/378: I suggest modifying the sentence “DAIR should be reconsidered”. With the results presented here, the authors may conclude that DAIR in those cases have more chances to fail (for instance, we do not know that if the chances of infection cure are higher if DAIR is reconsidered and a 2-stage exchange is performed)
Author Response
The authors thank the reviewers for carefully considering the manuscript and acknowledging our constructive engagement.
Point 1: I would suggest revising to a more “easy” English (for instance “complex constellation of interrelated factors”
Response 1: We appreciate the reviewer’s commentary and we have been modified the quoted sentence (page 2, line 63).
Point 2: Introduction. Line 44: refer to “faster recovery” or “better functional outcomes” since most of patients that need a prosthesis are retired.
Response 2: Given the cohort's age range, it is a good point. We appreciate the reviewer’s suggestion, and we will modify that sentence accordingly (page 1, line 44).
Point 3: M&M. How infection was ruled out in revisions?
Response 3: Particularly the DAIR procedure is carefully detailed in M&M (page 3, lines 129-137). Intraoperative fluid and tissue samples were taken for cultures in each of the 64 cases, with no difference between debridements performed on primary or revision cases.
Regarding the index surgery, a previous aspiration and blood reactants were not performed in all revisions because unfortunately in the past it was not common practice in our hospital. For example, preoperative joint aspiration was hardly conducted in those cases where the revision cause was obvious (and non septic) i.e. instability, mechanical issues, fracture…
Point 4: M&M. DAIR procedure include exchange of the non-fixed components. Why the surgeon decided in some cases not to revise them?
Response 4: We observed an evident increase in the percentage of non-fixed parts exchange from 0-20% in 2011 to 70% in 2018 in our cohort, as the importance of exchanging those mobile parts became more evident in recent years. More information on this issue and the significance of the study is provided in point 5.
Point 5: Results. Some 30% of cases had a prosthesis for another reason than OA. Which are those diagnoses?
Response 5: This is a good point, we considered to specify these diagnoses, however we decided to simplify as OA vs Non-OA due to our limited number of cases and the heterogeneity of those diagnoses .
Such diagnoses are Avascular necrosis, Hip displasia, Fracture and fracture related secuelae, mechanical issues, instability, aseptic loosening, patellofemoral issues etc…
Based on our experience, the study to be validated (Shohat) and previous related litterature, we believe this a good manner to classify our diagnosed patients.
Point 6: 50% of cases had not liner/ polyethylene exchange. This is my major concern of the present study as there is quite good evidence that it is a risk factor for DAIR failure. It should be stated as a limitation of the study.
Response 6: We appreciate the reviewer’s suggestion. Certainly a large body of research on the subject indicates that changing the modular components is beneficial to the outcome of the DAIR. However, over the past ten years, our approach in this area has been modified (as in many other institutions). We observed an evident increase in the percentage of poly exchange from 0-20% in 2011 to 70% in 2018 in our cohort, as the importance of exchanging those mobile parts became more evident in recent years. However, our mobile parts exchange rate (53%) is not significantly different from that reported by Shohat (49%) and therefore we do not believe it is necessarily a limitation of the study.
This leads to an open discission about how, despite a low rate of change of mobile parts, the failure rate of DAIR in our cohort (40%) is not very high, but it remains within those usually published? This issue is outlined in the discussion (page 10, line 386-391).
Point 7: Lines 377/378: I suggest modifying the sentence “DAIR should be reconsidered”. With the results presented here, the authors may conclude that DAIR in those cases have more chances to fail (for instance, we do not know that if the chances of infection cure are higher if DAIR is reconsidered and a 2-stage exchange is performed)
Response 7: We appreciate the reviewer’s suggestion. We have modified the paragraph as: “According to the findings from our study, performance of a DAIR procedure should be seriously meditated in patients with an acutely infected primary arthroplasty implanted for an indication different from OA or a revision arthroplasty, or in those with plasma CPR levels above 30mg/L”